# Morphology, phylogeny, and taxonomy of two species of colonial volvocine green algae from Lake Victoria, Tanzania

Hisayoshi Nozaki[1]*, Ryo Matsuzaki[2,3], Benedicto Boniphace Kashindye[4], Charles Nyarongo Ezekiel[4], Sophia Shaban[4], Masanobu Kawachi[2], Mitsuto Aibara[5], Masato Nikaido[5]

**1** Department of Biological Sciences, Graduate School of Science, The University of Tokyo, Tokyo, Japan, **2** Center for Environmental Biology and Ecosystem Studies, National Institute for Environmental Studies, Tsukuba, Ibaraki, Japan, **3** Faculty of Life and Environmental Sciences, University of Tsukuba, Tsukuba, Ibaraki, Japan, **4** Tanzania Fisheries Research Institute (TAFIRI), Mwanza, Tanzania, **5** School of Life Science and Technology, Tokyo Institute of Technology, Tokyo, Japan

* nozaki@bs.s.u-tokyo.ac.jp

## Abstract

The biodiversity and taxonomy of colonial volvocine green algae are important in ancient lakes in tropical regions. However, few taxonomic studies of these algae have been conducted in African ancient lakes. Here, we describe two species of colonial volvocine green algae in cultures originating from water samples from Lake Victoria, an ancient lake in Africa. One was identified as an undescribed morphological species of *Eudorina*; *E. compacta* sp. nov. This new species can be distinguished from other *Eudorina* species by its compactly arranged vegetative cells that form a hollow ellipsoidal colony. The other was identified as *Colemanosphaera charkowiensis*. The genus *Colemanosphaera* is new to Africa.

## Introduction

The volvocine green algae are a model lineage for studying the evolution of sex and multicellularity [1,2]. Culture materials established from samples we recently collected in Thailand contained two colonial volvocine species that are distributed only in a restricted tropical area: *Pleodorina sphaerica* [3] and *Volvox dissipatrix* [4]. Based on samples collected from Lake Biwa, a Japanese ancient lake, in 2013, *Volvox africanus* was recorded for the first time in Japan and a new species of *Volvox*, *V. reticuliferus*, was described [5]. Ancient lakes in the world provide significantly high diversity and levels of endemicity in animals and diatoms [6–8]. Therefore, field collections in African ancient lakes should be fruitful for studying the biodiversity and taxonomy of colonial volvocine green algae.

In Africa, several species of the genus *Volvox* have been recorded based on field samples collected since the 20[th] century [9–11]. Cultures of colonial volvocine green algae have been established from soil samples collected in Africa, including *Volvox capensis* [12], *V. rousseletii* [13], and *Gonium pectorale* [14]. However, colonial volvocine green algae have not been recorded from African ancient lakes.

**Data Availability Statement:** New sequence data, alignments used for our phylogenetic analyses, and new strains are available under the DDBJ/ENA/ GenBank accession numbers (LC504534–

LC504554), TreeBASE ID (S25155), and NIES Collection strain designations (NIES-4373–NIES-4380), respectively. All other relevant data are within the paper and its Supporting Information files.

**Funding:** This study was supported by a Grants-in-Aid for Scientific Research (grant numbers 16H02518 for HN and 17H04606 for MN) from the Ministry of Education, Culture, Sports, Science and Technology (MEXT)/Japan Society for the Promotion of Science (JSPS) KAKENHI (https://www.jsps.go.jp/english/e-grants/). The funders had no role in study design, data collection and analysis, decision to publish, or preparation of the manuscript.

**Competing interests:** The authors have declared that no competing interests exist.

In December 2018, we visited Lake Victoria, an ancient lake in Africa [6,8]. Based on material cultured from water samples collected from Lake Victoria, two colonial volvocine genera were identified: *Eudorina* and *Colemanosphaera*. This report describes the morphology, phylogeny, and taxonomy of these two algae.

## Materials and methods

### Ethics statement

We collected colonial volvocine green algae from the water column of Lake Victoria. S1 Table shows the collection locations and details. Research in Mwanza was permitted by the Tanzania Commission for Science and Technology (COSTECH) (Nos. 2018-525-NA-2018-213 and 2018-527-NA-2018-213 for HN and RM, respectively).

### Establishment of cultures and morphological observations

Water samples were collected from Nyegezi, Mwanza within the Mwanza Gulf, in southern Lake Victoria, 3–5 December 2018 (S1 Table). Using the pipette-washing method [15], 23 clonal cultures of colonial volvocine green algae (*Eudorina* and *Colemanosphaera*) were established from the water samples. The cultures were maintained in screw-cap tubes ($18 \times 150$ mm) containing 10–11 mL artificial freshwater-6 (AF-6) or Volvox thiamin acetate (VTAC) medium [16] at 25 °C with a 14:10 h light:dark schedule under cool-white fluorescent lamps at an intensity of 80–130 µmol $m^{-2}$ $s^{-1}$. Eight of these new wild strains were selected for detailed studies and are available from the Microbial Culture Collection at the National Institute for Environmental Studies [16; https://mcc.nies.go.jp/index_en.html] as NIES-4373– NIES-4380 (S1 Table).

Vegetative colonies and asexual reproduction of *Colemanosphaera* and *Eudorina* were observed by examining a small aliquot of colonies that were maintained by inoculating 0.5–1.0 mL of actively growing culture into fresh medium every 5–7 days.

To induce the production of sperm packets by *Eudorina* male strains (the gender was selected by genomic PCR as described below), an actively growing culture (10 mL VTAC medium at 25 °C as described above) was mixed with 20 mL of *Pleodorina* mating medium [17] in Petri dishes ($20 \times 90$ mm). Zygotes were induced by mixing male and female strains (10 mL in total) with 20 mL of *Pleodorina* mating medium in Petri dishes.

Light microscopy was performed using a BX60 microscope (Olympus, Tokyo, Japan) equipped with Nomarski interference optics. For transmission electron microscopy of *Eudorina*, colonies in culture were subjected to double fixation (using 1.5% glutaraldehyde for prefixation and 2% $OsO_4$ for postfixation) and examined as described previously [18] using a JEM-1010 electron microscope (JEOL, Tokyo, Japan).

### Molecular experiments

To determine the gender of the *Eudorina* strains, the presence or absence of the male-specific minus dominant gene (*MID*) was examined by genomic PCR with *MID*-specific primers (S2 Table). PCR was carried out as described previously [19].

Sequences of the internal transcribed spacers of nuclear ribosomal DNA (ITS rDNA [ITS-1, 5.8S rDNA, and ITS-2]) and chloroplast Rubisco large subunit genes (*rbcL*) of the strains were obtained as described previously [19]. To construct ITS rDNA and *rbcL* phylogenies, we analyzed the operational taxonomic units or species/samples/strains listed in S1 and S3 Tables. The sequences were aligned as described previously [20]. The alignments are available from TreeBASE (www.treebase.org/treebase-web/home.html; study ID: S25155). The root or

outgroup was designated based on previous phylogenetic analyses of colonial volvocine algae [20]. Maximum-likelihood (ML) analyses based on the ITS rDNA and *rbcL* alignments were performed using MEGA version X [21], with 1000 bootstrap replicates [22]. In addition, Bayesian phylogenetic analyses of the respective alignments were carried out using MrBayes 3.2.6 [23], as described previously [24]. The secondary structures of ITS-2 were predicted as described previously [20].

To calculate genetic distances, the ATP synthase subunit beta (*atpB*) and photosystem I P700 chlorophyll a apoprotein A2 (*psaB*) genes of a *Eudorina* strain (2018-1205-E14; S1 Table) and ITS rDNA of *Eudorina illinoisensis* strain NIES-460 were determined as described previously [19,25].

## Nomenclature

The electronic version of this article in Portable Document Format (PDF) in a work with an ISSN or ISBN constitutes a published work according to the International Code of Nomenclature for algae, fungi, and plants; hence, the new names contained in the electronic publication of a PLOS ONE article are effectively published under that Code from the electronic edition alone, so there is no longer any need to provide printed copies.

## Results and discussion

### Molecular analyses

Based on the sequences of ITS rDNA and the chloroplast *rbcL* gene, the strains were classified into two types: *Eudorina* and *Colemanosphaera*. The 661 (*Eudorina*) or 669 (*Colemanosphaera*) base-pair sequences of ITS rDNA and 1128 base pairs of *rbcL* were completely identical among the strains of each type (S1 Table). Fig 1 shows the phylogenetic positions of the two Tanzanian species within the Volvocaceae, based on the *rbcL* sequences.

In the *rbcL* gene phylogeny (Fig 1), the Tanzanian strains of *Eudorina compacta*, *E. minodii*, and several strains of *E. elegans* formed a clade with 64% bootstrap support in the ML tree and 1.00 posterior probability for the BI tree. The Tanzanian alga *E. compacta* was robustly sister to *E. elegans* strain NIES-456 originating from Tokyo, Japan [26,27]. In addition, the Tanzanian *Colemanosphaera* strains were robustly included within the genus *Colemanosphaera*, and were closely related to *C. charkowiensis* from Lake Isanuma, Japan [20].

The ITS-2 phylogenetic analyses produced a more detailed phylogeny of the Tanzanian algae. A Korean strain (KMMCC 1257) identified as "*Pandorina morum*" was almost identical to the Tanzanian strains of *Eudorina compacta* in the ITS-2 sequence (S1 Fig). Based on comparisons of the secondary structure of ITS-2, no compensatory base changes (CBCs) were found between the Tanzanian *Eudorina compacta* and Korean alga (S2 Fig). Two CBCs were found in the secondary structure of ITS-2 between *E. compacta* and *E. elegans* strain NIES-456 (S2 Fig). The Tanzanian strains of *Colemanosphaera* and Japanese strains *C. charkowiensis* formed a small robust clade in which no CBC was detected in the secondary structure of ITS-2 (S3 and S4 Figs).

### Morphological observations and taxonomy

***Eudorina compacta* sp. nov.**.    *Morphological observations*: Vegetative colonies were ellipsoidal or elongate-ovoid in shape, contained 16 or 32 cells of approximately identical size, and measured up to 95 μm long. The cells were compactly arranged at the periphery of the gelatinous matrix to form a hollow colonial structure (Fig 2A and 2B). The cells were hexagonal or pentagonal in the surface view due to mutual compression, generally had no fenestrations or

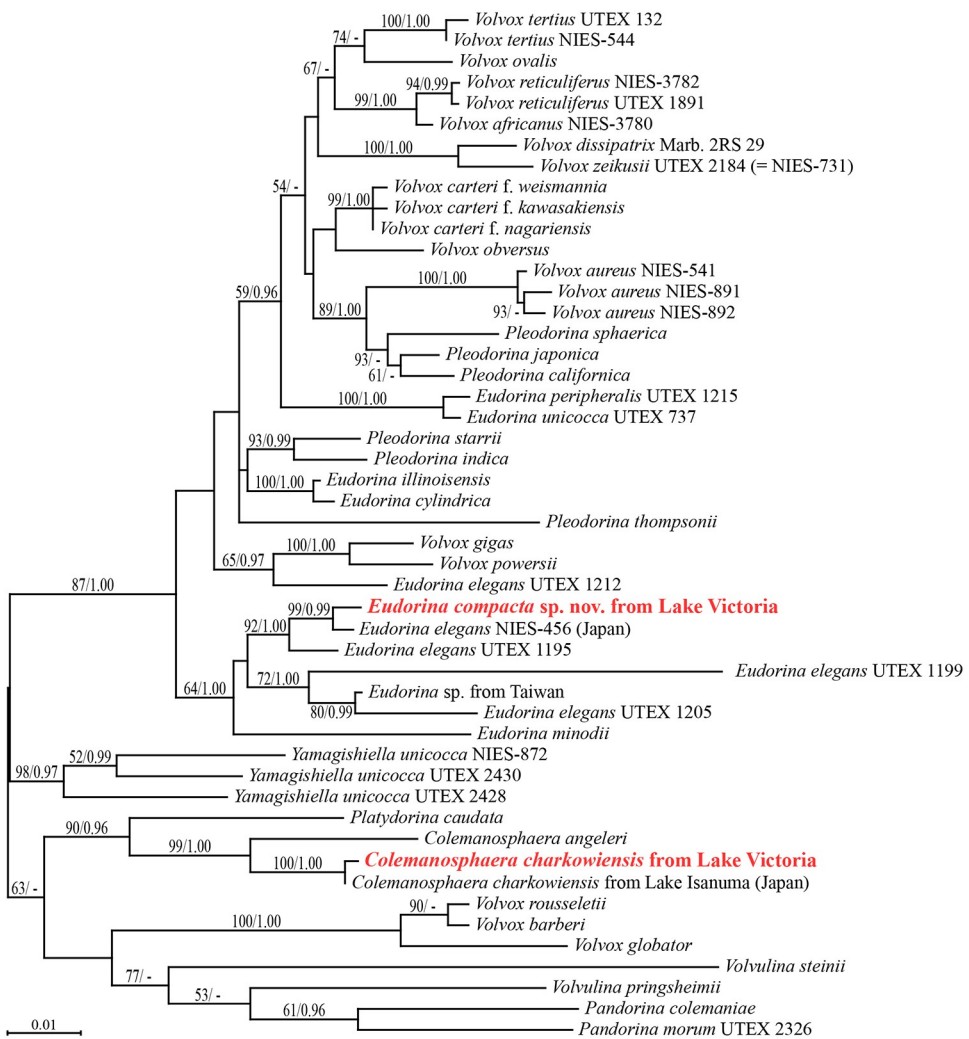

**Fig 1. Phylogenetic positions of two species of colonial volvocine green algae originating from Lake Victoria (red) within the Volvocaceae (S1 Table) based on maximum likelihood (ML) analysis of *rbcL* gene sequences.** Bootstrap values from ML (left) and posterior probabilities from Bayesian (right) analyses are shown on the branches.

spaces between them, and measured up to 23 μm in surface diameter (Fig 2C and 2D). Each cell was biflagellate and had a massive cup-shaped chloroplast with a stigma. Several or more contractile vacuoles were distributed throughout the surface of the protoplast of each cell (Fig 2C). There was a gradual decrease in stigma size from the anterior to posterior pole of the colony (Fig 2C). Three to seven pyrenoids of almost identical size were randomly distributed throughout the chloroplasts of mature vegetative cells (Fig 2D). All colonial cells performed asexual reproduction to form daughter colonies, as described previously in other species of *Eudorina* [28,29].

In sexual reproduction, each cell in male colonies divided successively to form 32-celled sperm packets (bundles of spindle-shaped male gametes) (Fig 2E). Zygotes were spherical in shape, with a smooth wall, and measured 15–18 μm in diameter (Fig 2F).

Under a transmission electron microscope, the entire colony was surrounded by a tripartite boundary (colonial boundary) of extracellular matrix (Fig 3), as in other volvoceans [18,20].

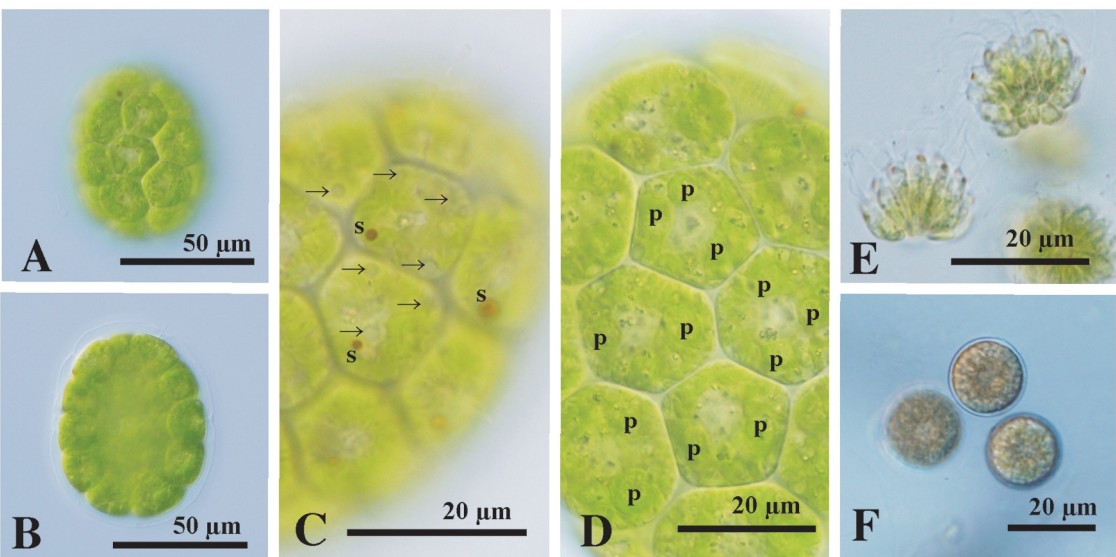

**Fig 2. Light microscopy of *Eudorina compacta* Nozaki sp. nov. originating from Lake Victoria.** (A)-(D) Vegetative colonies of strain 2018-1205-E14. (A) Surface view of 32-celled colony showing compactly arranged cells. (B) Optical section of 32-celled colony showing a hollow structure. (C) Surface view of colonial cells showing a stigma (s) and contractile vacuoles (arrows) randomly distributed on the cell surface. (D) Optical section of colonial cells with multiple pyrenoids (p) within the chloroplast. (E) Formation of sperm packets (bundles of male spindle-shaped male gametes. Strains 2018-1205-E8 and E14. (F) Mature zygotes. Strains 2018-1205- E14 and TzCl-9.

Each cell was enclosed tightly by a dense layer (cellular envelope) of matrix (Fig 3A–3C). The cellular envelopes of adjoining cells were tightly attached to one another in some sections (Fig 3B). The structure and arrangement of organelles within the protoplast (Fig 3C) were essentially the same as those in other volvocaceans [18].

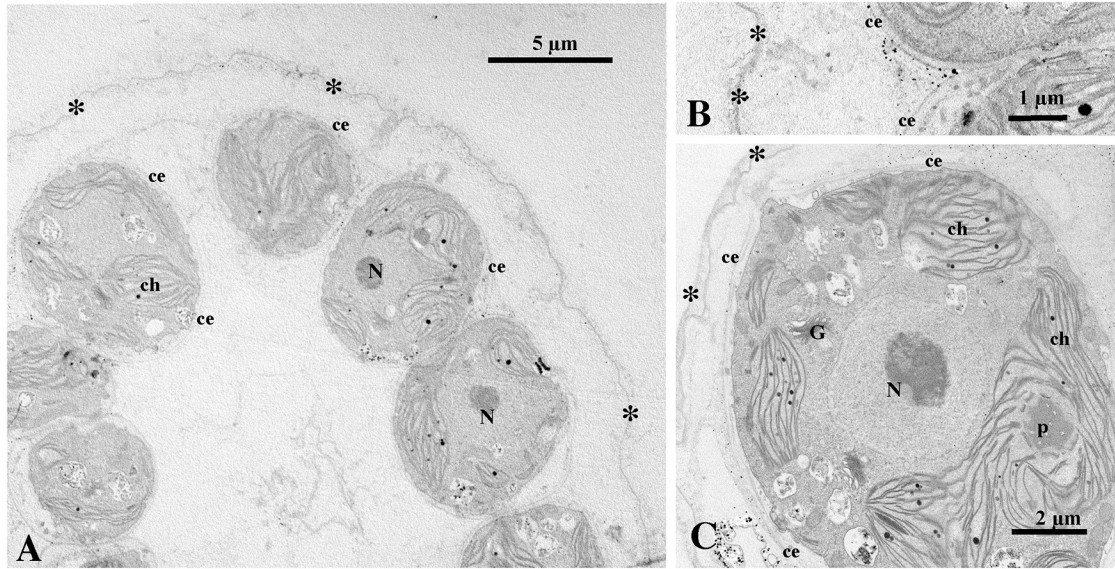

**Fig 3. Transmission electron microscopy of vegetative colonies of *Eudorina compacta* Nozaki sp. nov. strain 2018-1205- E14 originating from Lake Victoria.** Each cell is enclosed by a dense fibrillar layer (cellular envelope) of the extracellular matrix inside a tripartite layer or colonial boundary (asterisks) encompassing the whole colony. Abbreviations: ce, cellular envelope; ch, chloroplast; G, Golgi body; N, nucleus; p, pyrenoid. (A) Longitudinal section of colony showing hollow colonial structure. (B) Part of colony showing attachment of cellular envelopes of neighboring cells. (C) Longitudinal section of cell.

*Remarks*: Under a light microscope, the African alga *Eudorina compacta* is similar to *Pandorina morum* and *P. colemaniae* [30–32] in having cells arranged compactly to be angular in the surface view due to mutual compression. However, the ultrastructure of the extracellular matrix in *E. compacta* differs from that of *P. morum* and *P. colemaniae* [32,33]. In the *Pandorina* species, each cell of the colony lacks cellular envelopes inside the colonial boundary. By contrast, *E. compacta* has cellular envelopes tightly enclosing cells inside the colonial boundary (Fig 3) as in other volvocacean genera with spheroidal colonies without the differentiation of obligate somatic cells: *Yamagishiella*, *Eudorina*, and *Colemanosphaera* [18,20]. Of these three genera, *E. compacta* is morphologically consistent with *Eudorina* in having several or more contractile vacuoles throughout the surface of the vegetative cell (Fig 2C) and anisogamous sexual reproduction with the formation of sperm packets (Fig 2E) [20].

Morphologically, *E. compacta* is unique within the genus *Eudorina* in having cells compactly arranged to form a hollow spheroidal colony, generally without spaces between the adjoining cells in the surface view [27–31]. Our *rbcL* gene phylogeny clearly placed *E. compacta* in a robust clade composed of several strains of *Eudorina elegans* [28] and *E. minodii* [29] (Fig 1). Furthermore, this African species was sister to *Eudorina elegans* strain NIES-456 originating from Japan [32] and these two entities showed genetic differences in ITS-2 (the presence of two CBCs) and *rbcL* (7 of 1128 bp), *atpB* (10 of 1128 bp) and *psaB* (12 of 1494 bp) genes, which were sufficient to recognize different species of *Eudorina* (S1, S2 and S5–S7 Figs). Although *E. compacta* represents a derived lineage from *E. elegans sensu* Goldstein [28], the latter morphological species is apparently composed of multiple cryptic species, as suggested by its sexual isolation [28,34] and genetic diversity (Fig 1). Further detailed studies will construct a more natural taxonomic system for the species of *Eudorina* related to *E. compacta*.

**Colemanosphaera charkowiensis.** *Morphological observations*: Vegetative colonies were ovoid to ellipsoidal in shape, consisted of 16 or 32 cells of approximately identical sizes, and measured up to 78 μm long. The colonial cells were embedded at the periphery of the gelatinous matrix to form a hollow colonial structure (Fig 4A and 4B). The cells were ovoid to subspheroidal in shape, with a broad anterior face that was more or less angular, measuring up to 19 μm in surface diameter. Each cell was biflagellate and had a massive cup-shaped chloroplast with a stigma. Generally, two contractile vacuoles were located only near the base of the flagella of each cell (Fig 4C). There was a gradual decrease in stigma size from the anterior to posterior pole of the colony. The chloroplast contained strong longitudinal striations on the surface (Fig 4C and 4D). Three to six pyrenoids of almost identical size were distributed in the chloroplasts of mature vegetative cells (Fig 4D). All colonial cells performed asexual reproduction to form daughter colonies, as described previously [20].

*Remarks*: Within the colonial volvocine green algae, three genera (*Yamagishiella*, *Colemanosphaera*, and *Eudorina*) have 32-celled spheroidal colonies without somatic cell differentiation [20,31]. While *Eudorina* has multiple contractile vacuoles randomly distributed on the surface of each vegetative cell, *Yamagishiella* and *Colemanosphaera* have only anterior contractile vacuoles in each vegetative cell as in the African alga [20,35]. The latter two genera have different modes of sexual reproduction [20]. However, sexual reproduction was not observed in the present strains. The *rbcL* gene sequence data and strong longitudinal striations on the chloroplast surface and multiple pyrenoids of almost identical size in the chloroplast clearly assigned the present species to *C. charkowiensis* [20]. No such strong striations of the chloroplast surface are seen in the other species of *Colemanosphaera*, *C. angeleri*, which contains a large basal pyrenoid and small pyrenoids in the cup-shaped chloroplasts in mature vegetative cells [20]. *Yamagishiella* contains only a single basal pyrenoid in the chloroplast [35].

The genus *Colemanosphaera* was established based on *C. charkowiensis*, which was originally described as *Pandorina charkowiensis* originating from the Ukraine [20]. Nozaki *et al*.

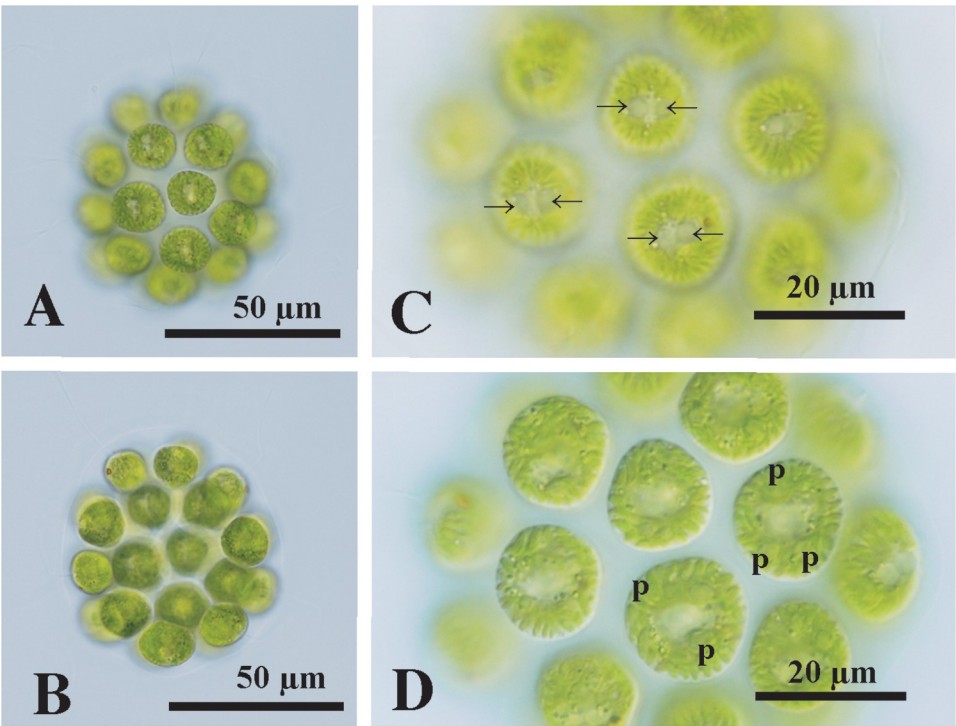

**Fig 4. Light microscopy of 32-celled vegetative colonies of *Colemanosphaera charkowiensis* strain 2018-1204-C1 originating from Lake Victoria.** (A), (B) Two views of colony. (C) Surface view of colonial cells showing anterior contractile vacuoles (arrows). (D) Optical section of colonial cells showing multiple pyrenoids (p) in the chloroplast.

[20] studied Japanese strains of *C. charkowiensis* and *C. angeleri* and analyzed the ITS region of *C. angeleri* from Austria. Recently, *C. angeleri* and *C. charkowiensis* were recorded from China [36,37]. Ours is the first record of the genus *Colemanosphaera* in Africa.

## Taxonomic treatment

*Eudorina compacta* **Nozaki sp. nov..** Vegetative colonies ellipsoidal or elongate-ovoid in shape, containing 16 or 32 cells of approximately identical size, measuring up to 95 μm long. Cells compactly arranged to form a hollow colonial structure. The cells hexagonal or pentagonal in the surface view due to mutual compression, measuring up to 23 μm in the surface diameter. Each cell biflagellate, having a massive cup-shaped chloroplast with three to seven pyrenoids of almost identical size and several or more contractile vacuoles distributed throughout the surface. All colonial cells performing daughter colony formation in asexual reproduction. Sexual reproduction anisogamous with formation of sperm packets (bundles of male gametes). Under the transmission electron microscope, the whole vegetative colony surrounded by a tripartite boundary (colonial boundary) of the extracellular matrix, and each cell enclosed tightly by a dense layer (cellular envelope) of the matrix.

Holotype: Resin-embedded asexual spheroids of *Eudorina compacta* strain 2018-1205-E14 (TNS-AL-58961), deposited in TNS (Department of Botany, National Museum of Nature and Science, Tsukuba, Japan). This strain is available as NIES-4373 from the Microbial Culture Collection at the National Institute for Environmental Studies, Japan [16].

Strains examined: 2018-1205-E14 (= NIES-4373), 2018-1205-E11 (= NIES-4374), 2018-1205-E8 (= NIES-4375), TzCl-9 (= NIES-4376) and TzCl-3 (= NIES-4377) (S1 Table).

Etymology: The species epithet "*compacta*" meaning compactness of vegetative cells.

Type locality: Lake Victoria, Mwanza, Tanzania. A water sample was collected by RM and HN on 5 December 2018.

## Supporting information

**S1 Fig. Maximum likelihood (ML) tree of *Eudorina* species related to *Eudorina compacta* from Lake Victoria based on ITS region of nuclear ribosomal DNA (ITS-1, 5.8S rDNA, and ITS-2) (S1 Table).**
(DOCX)

**S2 Fig. The secondary structure of nuclear ribosomal DNA (rDNA) internal transcribed spacer 2 (ITS-2) transcript of *Eudorina compacta* from Lake Victoria, including the 3' end of the 5.8S ribosomal RNA (rRNA) and the 5' end of the LSU rRNA.**
(DOCX)

**S3 Fig. Maximum likelihood (ML) tree of *Colemanosphaera* strains including those originating from Lake Victoria based on ITS region of nuclear ribosomal DNA (ITS-1, 5.8S rDNA, and ITS-2) (S1 Table).**
(DOCX)

**S4 Fig. The secondary structure of nuclear ribosomal DNA (rDNA) internal transcribed spacer 2 (ITS-2) transcript of *Colemanosphaera charkowiensis* from Lake Victoria, including the 3' end of the 5.8S ribosomal RNA (rRNA) and the 5' end of the LSU rRNA.**
(DOCX)

**S5 Fig. Comparison of genetic distances between closely related strains in the genus *Eudorina*.**
(DOCX)

**S6 Fig. The secondary structure of nuclear ribosomal DNA (rDNA) internal transcribed spacer 2 (ITS-2) transcript of *Eudorina peripheralis* strain UTEX 1215, including the 3' end of the 5.8S ribosomal RNA.**
(DOCX)

**S7 Fig. The secondary structure of nuclear ribosomal DNA (rDNA) internal transcribed spacer 2 (ITS-2) transcript of *Eudorina cylindrica* strain UTEX 1197, including the 3' end of the 5.8S ribosomal RNA (RNA) and the 5' end of the LSU rRNA.**
(DOCX)

**S1 Table. List of colonial volvocine species/strains included in the phylogenetic analyses of ITS-2 sequences (with DDBJ/EMBL/GENBANK accession numbers; S1 and S3 Figs).**
(DOCX)

**S2 Table. Primers used for genomic PCR of possible male-specific minus dominance (*MID*) gene of *Eudorina compacta*.**
(DOCX)

**S3 Table. List of the colonial volvocine taxa/strains included in the phylogenetic analysis (Fig 1) and DDBJ/EMBL/GenBank accession numbers of *rbcL* genes.**
(DOC)

## Acknowledgments

The authors acknowledge Dr. Semvua Isa Mzighani, Dr. Magreth Jackton Musiba and Mr. Hillary Deogratias John Mrosso from Tanzania Fisheries Research Institute (TAFIRI) who helped in the field survey arrangement and data collection.

## Author Contributions

**Conceptualization:** Hisayoshi Nozaki, Ryo Matsuzaki.

**Data curation:** Hisayoshi Nozaki, Ryo Matsuzaki.

**Formal analysis:** Hisayoshi Nozaki, Ryo Matsuzaki.

**Funding acquisition:** Hisayoshi Nozaki, Masato Nikaido.

**Investigation:** Hisayoshi Nozaki, Ryo Matsuzaki, Mitsuto Aibara.

**Methodology:** Hisayoshi Nozaki, Ryo Matsuzaki, Charles Nyarongo Ezekiel.

**Project administration:** Hisayoshi Nozaki, Benedicto Boniphace Kashindye, Mitsuto Aibara, Masato Nikaido.

**Resources:** Hisayoshi Nozaki, Charles Nyarongo Ezekiel, Masanobu Kawachi, Mitsuto Aibara.

**Supervision:** Hisayoshi Nozaki, Benedicto Boniphace Kashindye, Charles Nyarongo Ezekiel, Sophia Shaban, Masanobu Kawachi, Mitsuto Aibara, Masato Nikaido.

**Validation:** Hisayoshi Nozaki, Ryo Matsuzaki.

**Visualization:** Hisayoshi Nozaki, Ryo Matsuzaki.

**Writing – original draft:** Hisayoshi Nozaki, Ryo Matsuzaki.

**Writing – review & editing:** Hisayoshi Nozaki, Ryo Matsuzaki, Benedicto Boniphace Kashindye, Charles Nyarongo Ezekiel, Sophia Shaban, Masanobu Kawachi, Mitsuto Aibara, Masato Nikaido.

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
