## [Decision Letter · Decision Letter 0]

2 Oct 2019

PONE-D-19-26214

Morphology, phylogeny, and taxonomy of two species of colonial volvocine green algae from Lake Victoria, Tanzania

PLOS ONE

Dear Dr. Nozaki,

Thank you for submitting your manuscript to PLOS ONE. After careful consideration, we feel that it has merit but does not fully meet PLOS ONE’s publication criteria as it currently stands. Therefore, we invite you to submit a revised version of the manuscript that addresses the points raised during the review process.

I am requesting some minor changes as described below.

In your revision please consider the suggestions of Reviewer 1 including the request for inclusion of more information on biodiversity and geographical ranges.  The significance of the collection site being an ancient lake (versus another type of lake or aquatic habitat) was not clear, so if you leave the “ancient lake” description in the manuscript please add some additional information to provide context on why this is an important distinction to make.

For Reviewer 2, Point 1, PLoS ONE does not discriminate based on perceived impact or target audience size, so no need to address this comment.  Point 2 raises a valid issue that continues to complicate taxonomic classification of volvocine algae due to extensive polyphyly, but I don’t feel it is something that must be solved here in order to publish this work.

We would appreciate receiving your revised manuscript by Nov 16 2019 11:59PM. To enhance the reproducibility of your results, we recommend that if applicable you deposit your laboratory protocols in protocols.io, where a protocol can be assigned its own identifier (DOI) such that it can be cited independently in the future. For instructions see: http://journals.plos.org/plosone/s/submission-guidelines#loc-laboratory-protocols

We look forward to receiving your revised manuscript.

Kind regards,

James G. Umen, Ph. D.

Academic Editor

PLOS ONE

**Journal Requirements:**

**Comments to the Author**

1. Is the manuscript technically sound, and do the data support the conclusions?

Reviewer #1: Yes

Reviewer #2: Yes

2. Has the statistical analysis been performed appropriately and rigorously? 

Reviewer #1: Yes

Reviewer #2: Yes

3. Have the authors made all data underlying the findings in their manuscript fully available?

Reviewer #1: Yes

Reviewer #2: Yes

4. Is the manuscript presented in an intelligible fashion and written in standard English?

Reviewer #1: Yes

Reviewer #2: Yes

5. Review Comments to the Author

Reviewer #1: This manuscript describes the isolation of two volvocine green algal species from Lake Victoria, a novel species, Eudorina compacta, and the first isolation of Colemanosphaera charkowiensis in Africa. The manuscript presents novel research, which is worthy of publication, and is clearly and concisely written. The methods are rigorous and the images are excellent. I have one suggestion meant to improve the manuscript- both the abstract and introduction overemphasize that Lake Victoria is an "ancient" lake. The introduction would be more convincing if the emphasis on ancient lakes was reduced and the value of characterizing species ranges and cataloguing global/African biodiversity was emphasized more. Similarly, the previously known species range of C. charkowiensis should be discussed in the Introduction.

Minor comment:

Page 5, second to last line of "Molecular analyses" section: "stains" should be "strains".

Reviewer #2: In the submitted manuscript, the authors described two colonial volvocalean green algae isolated from Lake Victoria (Tanzania). The ms is well written and all results are sound. The phylogenetic analyses are state-of-the-art. However, I have two major concerns.

1. The story is very small and is only of interests for small community. Therefore, I don't know if that should be published in a journal of wider audience such as PLOS One, but I leave this point to the editor.

2. Many studies about volvocalean algae have demonstrated that most genera are not monophyletic. So, the genus Eudorina is split at least into three lineages as the authors also showed. Therefore, in my view it is not possible to describe a new species of Eudorina if the status of this genus is not resolved. Even the type species of Eudorina, E. elegans, appears on different position in the phylogenetic analyses. It is clear that the new strain represents an own species, but if the authors want to describe a new species, the status of Eudorina elegans and the generic concept of Eudorina should be resolved. This can be done if the authors decide which strain of E. elegans will be designated as epitype. The epitype strain should be an isolate from Germany, near Berlin (type locality).

6. PLOS authors have the option to publish the peer review history of their article (what does this mean?). If published, this will include your full peer review and any attached files.

Reviewer #1: No

Reviewer #2: No

---

## [Author Response · Author response to Decision Letter 0]

8 Oct 2019

Dear Dr. Jim Umen,

 Thank you very much for your positive review results of our manuscript entitled "Morphology, phylogeny and taxonomy of two species of colonial volvocine green algae from Lake Victoria, Tanzania " submitted for publication in PLOS ONE as a research article.

 Based on the comments raised by Reviewers the academic editor, the manuscript has been revised adequately. Accession numbers of the new sequences, study ID of the rDNA ITS regions and rbcL-psbC sequences used for construction of the phylogenetic tress (Fig. 1 and S1 and S3 Figs) and NIES strain designations of the new Tanzanian strains of Eudorina compacta and Colemanosphaera charkowiensis have been obtained and described in the revised manuscript.

Our responses to the comments by the academic editor and Reviewers have been described below.

******

PONE-D-19-26214

Morphology, phylogeny, and taxonomy of two species of colonial volvocine green algae from Lake Victoria, Tanzania

PLOS ONE

Dear Dr. Nozaki,

Thank you for submitting your manuscript to PLOS ONE. After careful consideration, we feel that it has merit but does not fully meet PLOS ONE’s publication criteria as it currently stands. Therefore, we invite you to submit a revised version of the manuscript that addresses the points raised during the review process.

I am requesting some minor changes as described below.

In your revision please consider the suggestions of Reviewer 1 including the request for inclusion of more information on biodiversity and geographical ranges. The significance of the collection site being an ancient lake (versus another type of lake or aquatic habitat) was not clear, so if you leave the “ancient lake” description in the manuscript please add some additional information to provide context on why this is an important distinction to make.

Response: In order to explain the significance of the collection site being an ancient lake, the following three references ([6-8]) suggesting the importance of ancient lakes worldwide in the biodiversity studies have been cited in the Introduction section of the revised manuscript. 

6. Martens K. Speciation in ancient lakes. Trends Ecol Evol. 1997; 12: 177–182. doi: 10.1016/s0169-5347(97)01039-2.

7. Cristescu ME, Adamowicz SJ, Vaillant JJ, Haffner DG. Ancient lakes revisited: from the ecology to the genetics of speciation. Mol Ecol. 2010; 19: 4837–4851. doi: 10.1111/j.1365-294X.2010.04832.x.

8. Kulikovskiy MS, Lange-Bertalot H, Kuznetsova IV. Lake Baikal: hotspot of endemic diatoms II. In: Lange-Bertalot H. editor. Iconographia Diatomologica Volume: 26. Oberreifenberg: Koeltz Scientific Books; 2015. pp. 1-656. 

For Reviewer 2, Point 1, PLoS ONE does not discriminate based on perceived impact or target audience size, so no need to address this comment. Point 2 raises a valid issue that continues to complicate taxonomic classification of volvocine algae due to extensive polyphyly, but I don’t feel it is something that must be solved here in order to publish this work.

Response: I agree with these AE comments. See Responses described below.

We would appreciate receiving your revised manuscript by Nov 16 2019 11:59PM. To enhance the reproducibility of your results, we recommend that if applicable you deposit your laboratory protocols in protocols.io, where a protocol can be assigned its own identifier (DOI) such that it can be cited independently in the future. For instructions see: http://journals.plos.org/plosone/s/submission-guidelines#loc-laboratory-protocols

• A rebuttal letter that responds to each point raised by the academic editor and reviewer(s). This letter should be uploaded as separate file and labeled 'Response to Reviewers'.

• A marked-up copy of your manuscript that highlights changes made to the original version. This file should be uploaded as separate file and labeled 'Revised Manuscript with Track Changes'.

• An unmarked version of your revised paper without tracked changes. This file should be uploaded as separate file and labeled 'Manuscript'.

We look forward to receiving your revised manuscript.

Kind regards,

James G. Umen, Ph. D.

Academic Editor

PLOS ONE

Journal Requirements:

Comments to the Author

1. Is the manuscript technically sound, and do the data support the conclusions?

Reviewer #1: Yes

Reviewer #2: Yes

2. Has the statistical analysis been performed appropriately and rigorously? 

Reviewer #1: Yes

Reviewer #2: Yes

3. Have the authors made all data underlying the findings in their manuscript fully available?

Reviewer #1: Yes

Reviewer #2: Yes

4. Is the manuscript presented in an intelligible fashion and written in standard English?

Reviewer #1: Yes

Reviewer #2: Yes

5. Review Comments to the Author

Reviewer #1: This manuscript describes the isolation of two volvocine green algal species from Lake Victoria, a novel species, Eudorina compacta, and the first isolation of Colemanosphaera charkowiensis in Africa. The manuscript presents novel research, which is worthy of publication, and is clearly and concisely written. The methods are rigorous and the images are excellent. I have one suggestion meant to improve the manuscript- both the abstract and introduction overemphasize that Lake Victoria is an "ancient" lake. The introduction would be more convincing if the emphasis on ancient lakes was reduced and the value of characterizing species ranges and cataloguing global/African biodiversity was emphasized more. Similarly, the previously known species range of C. charkowiensis should be discussed in the Introduction.

Responses: In order to explain the significance of the collection site being an ancient lake, the following three references ([6-8]) suggesting the importance of ancient lakes worldwide in the biodiversity studies have been cited in the Introduction section of the revised manuscript. 

6. Martens K. Speciation in ancient lakes. Trends Ecol Evol. 1997; 12: 177–182. doi: 10.1016/s0169-5347(97)01039-2.

7. Cristescu ME, Adamowicz SJ, Vaillant JJ, Haffner DG. Ancient lakes revisited: from the ecology to the genetics of speciation. Mol Ecol. 2010; 19: 4837–4851. doi: 10.1111/j.1365-294X.2010.04832.x.

8. Kulikovskiy MS, Lange-Bertalot H, Kuznetsova IV. Lake Baikal: hotspot of endemic diatoms II. In: Lange-Bertalot H. editor. Iconographia Diatomologica Volume: 26. Oberreifenberg: Koeltz Scientific Books; 2015. pp. 1-656. 

Species range of C. charkowiensis had been discussed in the Results and Discussion section; it is not needed to discuss in the Introduction section.

Minor comment:

Page 5, second to last line of "Molecular analyses" section: "stains" should be "strains".

Response: Done as suggested.

Reviewer #2: In the submitted manuscript, the authors described two colonial volvocalean green algae isolated from Lake Victoria (Tanzania). The ms is well written and all results are sound. The phylogenetic analyses are state-of-the-art. However, I have two major concerns.

1. The story is very small and is only of interests for small community. Therefore, I don't know if that should be published in a journal of wider audience such as PLOS One, but I leave this point to the editor.

Response: PLoS ONE does not discriminate based on perceived impact or target audience size, so no need to address this comment. 

2. Many studies about volvocalean algae have demonstrated that most genera are not monophyletic. So, the genus Eudorina is split at least into three lineages as the authors also showed. Therefore, in my view it is not possible to describe a new species of Eudorina if the status of this genus is not resolved. Even the type species of Eudorina, E. elegans, appears on different position in the phylogenetic analyses. It is clear that the new strain represents an own species, but if the authors want to describe a new species, the status of Eudorina elegans and the generic concept of Eudorina should be resolved. This can be done if the authors decide which strain of E. elegans will be designated as epitype. The epitype strain should be an isolate from Germany, near Berlin (type locality).

Response: The main object of the present study is to delineate a new species of Eudorina. Although the polyphyly of the genus Eudorina has been a serious taxonomic problem, many further data of sequences and unobserved morphology are needed. Thus, the generic concept of the genus Eudorina has not been revised in this manuscript.

6. PLOS authors have the option to publish the peer review history of their article (what does this mean?). If published, this will include your full peer review and any attached files.

Do you want your identity to be public for this peer review? For information about this choice, including consent withdrawal, please see our Privacy Policy.

Reviewer #1: No

Reviewer #2: No

---

## [Editor Report · Decision Letter 1]

10 Oct 2019

Morphology, phylogeny, and taxonomy of two species of colonial volvocine green algae from Lake Victoria, Tanzania

PONE-D-19-26214R1

Dear Dr. Nozaki,

We are pleased to inform you that your manuscript has been judged scientifically suitable for publication and will be formally accepted for publication once it complies with all outstanding technical requirements.

With kind regards,

James G. Umen, Ph. D.

Academic Editor

PLOS ONE
---

## [Editor Report · Acceptance letter]

14 Oct 2019

PONE-D-19-26214R1 

Morphology, phylogeny, and taxonomy of two species of colonial volvocine green algae from Lake Victoria, Tanzania 

Dear Dr. Nozaki:

I am pleased to inform you that your manuscript has been deemed suitable for publication in PLOS ONE. Congratulations! Your manuscript is now with our production department. 

With kind regards,

on behalf of

Dr. James G. Umen 

Academic Editor

PLOS ONE